# Bacterial Contamination of Equine Dentistry Equipment—Effect of Cleaning and Disinfection

**DOI:** 10.3390/ani11082320

**Published:** 2021-08-05

**Authors:** Todd Alsing-Johansson, Anja Pedersen, Karin Bergström, Susanna Sternberg-Lewerin, Johanna Penell, Anna Bergh

**Affiliations:** 1Department of Clinical Sciences, Swedish University of Agricultural Sciences, 750 07 Uppsala, Sweden; anja.pedersen@slu.se (A.P.); johanna.penell@slu.se (J.P.); Anna.Bergh@slu.se (A.B.); 2Department of Animal Health and Antimicrobial Strategies, National Veterinary Institute, 751 89 Uppsala, Sweden; karin.bergstrom@sva.se; 3Department of Biomedical Sciences and Veterinary Public Health, Swedish University of Agricultural Sciences, 750 07 Uppsala, Sweden; susanna.sternberg-lewerin@slu.se

**Keywords:** infection prevention and control, biosecurity, contamination, dental handpiece

## Abstract

**Simple Summary:**

Some of the equipment used in equine dentistry is difficult to clean and disinfect. Since it is vital to avoid the spread of infections in equine healthcare it is important to develop practical and easy-to-follow methods for cleaning and disinfecting dental equipment. The aim of this study was to investigate hygiene in equine dentistry. Dental equipment and the head support, where horses rest their head during dental care, were sampled for the amount of bacteria between each patient before and after dental care as well as after cleaning and/or disinfecting. The amount of bacteria was, in general, high on dental equipment and the head support after dental procedures. Bacteria were found in different amounts on most of the dental equipment after cleaning or disinfecting, which indicates a risk for spreading infections when using the equipment. For the head support, cleaning and/or disinfecting generally resulted in a reduced amount of bacteria, indicating a lowered risk for spreading infections. There is a great need for evidence-based guidelines on hygiene in equine dentistry to decrease the risk of transmitting infections between patients, facilities, and stables.

**Abstract:**

Equine dentistry has developed immensely and human dental equipment, such as handpieces, are often used. Measures to avoid the spread of infectious microorganisms are important, but this is challenging since handpieces are difficult to decontaminate. Thus, it is necessary to develop effective IPC measures in equine dentistry. The aim of this study was to contribute to the evidence needed for future evidence-based guidelines on IPC by investigating hygiene in equine dentistry. Used handpieces and dummies (i.e., handpieces not used during dental procedure, reflecting environmental bacterial contamination) and the head support were sampled each day before the first patient, for each patient after treatment, and after decontamination. All equipment was sampled with 3M ^TM^ Swab Samplers and the head support additionally sampled with dip slides. After dental procedures, the detected bacterial load was often high on used handpieces, dummies, and the head support. After decontamination, handpieces did not meet the criteria for high-level disinfected equipment. In all but one case decontamination of the head support resulted in a lowered bacterial load. There is a great need for evidence-based guidelines on hygiene in equine dentistry, including IPC measures, to decrease the risk of spreading infectious microorganisms between patients, facilities, and stables.

## 1. Introduction

Equine dental care is carried out at veterinary hospitals, clinics, and mobile practices. Equine dental health is a rapidly growing area in veterinary clinical practice. The frequency of treatments and types of dental procedures have increased during the last decades. The advancement of equine dental care in Sweden has made it common to use human dental equipment both for routine and advanced procedures. Handpieces, e.g., low-speed handpieces (LSH), surgical low-speed handpieces (SH), and high-speed handpieces (HSH), are used for both simple procedures, such as decreasing enamel ridges, and more advanced procedures, such as endodontic treatments. Even though protocols for cleaning, disinfecting, and sterilizing handpieces are in place for use in human dentistry there is a lack of knowledge regarding the cleaning, disinfecting, and sterilizing routines needed to ensure low risk for spreading infectious microorganisms between equine patients.

Measures to avoid the spread of infectious microorganisms between patients, and to staff, is common clinical practice in both human and veterinary medicine. This aspect of equine dentistry is challenging as handpieces are difficult to decontaminate, especially in mobile practices. Thus, there is a great need for effective infection prevention and control (IPC) measures in equine dentistry. The advanced dental procedures also entail a risk for transmission of infectious microorganisms between equine patients. For example, in a North American study, respiratory pathogens such as Equine herpes virus 1 and 4 (EHV-1, EHV-4), Equine influenza virus (EIV), Equine rhinitis B virus (ERBV), and/or methicillin-resistant *Staphylococcus aureus* (MRSA) were detected in 22% of healthy horses presented for routine dental care [1]. Furthermore, bacterial contamination of external and internal surfaces of handpieces has been shown after human dental procedures [2,3,4,5]. In addition, a higher degree of contamination of the environment and, thereby, an increased risk for patients when using high-speed devices, compared to using low-speed devices, has been shown [6,7]. There are, to our knowledge, no studies on contamination of handpieces or the surrounding environment in equine dentistry.

Based on use and hygiene, requirements for medical equipment, including dental handpieces, are categorized as non-critical, semi-critical, and critical [8]. The different categories require different levels of cleaning, disinfecting, and sterilizing (see Table 1). Handpieces are classified as semi-critical equipment, as long as they are used for non-sterile procedures, under Swedish and US hygiene guidelines for human dentistry as well as Swedish guidelines on IPC in equine healthcare and in small animal dentistry literature [8,9,10,11]. There is, however, no international consensus on criteria for the expected cleanliness for high-level disinfected equipment. Guidelines in human dentistry on how to clean, disinfect, and sterilize handpieces differ between countries. Several studies on human dental equipment demonstrate the challenges to meet criteria for both high-level disinfected and sterile equipment for handpieces. Surface disinfection of the external surface of handpieces resulted in failure to meet the criteria for high-level disinfection [12]. In another study, one of four cleaning devices intended for handpieces, the washer-disinfector (WD), provided an acceptable test result [13]. In one study, type N steam sterilizers failed to provide sterile handpieces whilst type B steam sterilizers provided sterile handpieces [14]. 

According to Pusterla et al. [1], equine dental equipment used for routine dental procedures, not expected to expose the pulp (i.e., semi-critical equipment), is rarely cleaned and disinfected between patients. Even though the equipment used for routine dental procedures can differ between countries, the conditions for cleaning procedures and risk of spreading infections can be assumed to be comparable. Moreover, IPC routines for equipment used in equine dentistry are not listed in the syllabus of equine dentistry courses in Sweden and in the European specialist program in equine dentistry [17,18]. In our experience, a commonly used decontamination method for handpieces in equine dentistry is surface disinfection with an intermediate disinfectant (a disinfectant with effect on most vegetative bacteria, some mycobacteria, some fungi, some enveloped and non-enveloped viruses [19]) without previous cleaning. However, no decontamination procedure in equine dentistry has yet been evaluated or published. Guidelines for IPC procedures in equine dental practice, based on solid data, are needed. 

The overall aim of this study was to contribute to the evidence needed for future guidelines on IPC procedures for equine dentistry by investigating hygiene in equine dentistry, specifically by: (1)assessing the bacterial load on handpieces and the patient environment during equine dental care;(2)assessing if manual cleaning of handpieces with detergent or disinfection with surface disinfection is sufficient to meet the Swedish criteria for high-level disinfected equipment; and(3)determining the bacterial load on the immediate surroundings after surface disinfection, or cleaning followed by surface disinfection.

## 2. Materials and Methods

The study was carried out in the dental practice of a veterinary hospital during the autumn of 2020. The veterinary hospital’s patient load consists of approximately 9000 patients per year and approximately 500 of them are dental patients. Sampling was carried out during two working days, with a total of 11 horses submitted for dental care.

### 2.1. Sampling

Equipment and surfaces to be sampled for bacterial load were selected based on a pilot study carried out in the dental practice of another veterinary hospital (for details from the pilot study see Appendix A: Bacterial load pilot study). The veterinary hospital’s patient load consists of approximately 5000 patients per year and approximately 500 of them are dental patients. Based on the pilot results, sampling of handpieces and the head support was standardized (for details about sampling methods tested, see Appendix A: Sampling methods tested). The sampling surfaces of the equipment are illustrated in Figure 1 and the sampling protocol is illustrated in Figure 2. Make and model of used handpieces can be found in the figure text in Figure 2. 

Handpieces were sampled with 3M ^TM^ Swab Samplers (3M ^TM^ Swab Sampler, Saint Paul, Minnesota, USA) with letheen broth, an already established method in the food industry for control of hygiene [20]. The 3M ™ Swab Samplers Method was modified to enable sampling of the different surfaces LSH, SH, and HSH, see Table 2 [21]. All samplings with 3M ™ Swab Samplers were carried out by the second author and the first author held the handpieces, wearing non-sterile nitrile gloves, during sampling. The samples were then analyzed for total aerobic colony count (ACC).

The samples from the head support were also analyzed for total ACC, using two sampling methods; 3M ^TM^ Swab Samplers with letheen broth (both study days) and dip slides (Envirocheck ^®^ Dip Slide Disinfection Control (DC), 9,4 cm^2^, Orange, USA) with TSA agar/TSA agar with a neutralizer, neutralizing several disinfectants (one of the study days). A particular 10 × 10 cm surface of the head support, a cushioned device supporting part of the horse’s lower jaw, was repeatedly swabbed at each sampling with the 3M ™ Swab Samplers Method, see Table 2 [21]. Also, an adjacent specified surface of the head support was sampled with dip slides. The surface chosen for sampling can be contaminated by fluid from the mouth during dental examination and treatment. The dip slide was pressed firmly to the surface for 15 s, then turned over and the opposite side of the slide was pressed against the adjacent surface for another 15 s [22]. All sampling with dip slides was carried out by the first author. 

For each sampling day and sampling method, one or two negative controls (unexposed dip slide and swab sampler) were applied. The controls were put in the dental care room just before sampling started for the day, approximately 10 min before the day’s first patient, and the controls were stored in the room until the gathering of sampling material after the last sampling of the day.

### 2.2. Cleaning and Disinfecting Methods

Five protocols for cleaning and disinfecting handpieces and the head support were used (for details see Table 3). Protocols 1 and 2 consisted of the veterinary hospital’s own protocol for surface disinfection with an intermediate disinfectant of handpieces and the head support. Protocol 3 consisted of manual cleaning of LSH and HSH and protocol 4 consisted of manual cleaning of SH. Protocol 5 consisted of manual cleaning of the head support followed by surface disinfecting with an intermediate disinfectant. LSH and SH were lubricated after every cleaning or disinfecting with a lubricant (PANA SPRAY Plus, NSK / Nakanishi inc., Kanuma, Japan) containing ester oil, ethanol butane, and propane. The HSH was lubricated using lubricating oil (MD-30 Advantage Dental Handpiece Oil MD-30, iM3, Sidney, Australia) containing synthetic hydrocarbon oil and ester oil after the last patient of the day. All handpieces were put back, out in the open, on a metal tray adjacent to the patient after cleaning or disinfecting and lubricating. The veterinary technician and the first author carried out every other cleaning or disinfection and lubrication of handpieces and the head support. During the period of sampling the dental care room and dental equipment were used three days a week. When protocols 1 and 2 were used, the dental care room and equipment had been unused for four days and when protocols 3–5 were used the dental care room and the dental equipment were used the day before. 

### 2.3. Bacteriological Analyses

All 3M Swab Samplers and dip slides were taken to the laboratory at the Swedish University of Agricultural Sciences on the day of sampling. 3M™ Swab Samplers were vortexed and 1 mL broth was drawn from the sampling tube and put onto a 3M Petrifilm ^TM^ aerobic count (AC) Plate (3M Petrifilm^TM^ Aerobic Count Plate, Saint Paul, Minnesota, USA), see Appendix A: Petrifilm ^TM^ negative control, as further described in the 3M ^TM^ Swab Sampler Method [21]. Samples were incubated aerobically in 30 ± 1 °C for 48 ± 2 h. Dip slides were incubated in 37 ± 1 °C for 48 ± 2 h. The colonies were counted manually by the first author, as described in the interpretation guide [23]. All 3M Petrifilm ^TM^ AC Plates and dip slides were photographed for documentation. 

### 2.4. Data Management

Microsoft ^®^ Excel ^®^ 2016 (16.0.5134.1000) (Microsoft Corporation, Redmond, Washington, USA) was used for data management and descriptive statistics. 

## 3. Results

The study included 80 samples from handpieces, 42 samples from dummies, 24 samples from the head support, and two negative controls using 3M ^TM^ Swab Samplers. In addition, dip slides were used in 14 samples from the head support and for two negative controls. No bacterial growth was detected in the samples from negative controls.

### 3.1. Handpieces

Bacterial growth was detected in all samples from HSH, both after dental procedures and after cleaning or disinfecting (see Table 4). In all samples from external surfaces from LSH and SH, bacterial growth was found after dental procedures, and after cleaning or disinfecting bacterial growth was still detected in all but one sample. After dental procedures, bacterial growth was detected in all samples from the coupling of LSH and in 6 of 10 samples from the shaft in SH. After cleaning or disinfecting, no bacterial growth was found in couplings from LSH while bacterial growth was found in 2 of 10 samples from the shaft of SH. 

### 3.2. Dummies

Bacterial growth was detected in all samples from external surfaces of dummies before the first patient of the day and in 11 of 12 samples from the external surface of dummies after dental procedures compared to 5 of 11 after cleaning or disinfection (see Table 5). In 2 of 3 samples from couplings of dummies, bacterial growth was detected before the first patient of the day. In addition, bacterial growth was detected in 2 of 11 samples from the coupling of dummies after cleaning or disinfecting. 

### 3.3. Head Support

In both samples from the head support before the first patient of the day bacterial growth was detected, and in one of them the CFU/cm^2^ were too numerous to count. The bacterial load detected on the head support was high (usually too numerous to count) after dental procedures. Both cleaning and/or disinfection reduced the bacterial load as seen in most of the samples (see Table 6). 

## 4. Discussion

To the best of our knowledge, this is the first study of hygiene in equine dentistry. According to Swedish guidelines in human dentistry, all handpieces should meet criteria for high-level disinfected equipment, and handpieces used for surgical procedures should be sterile [9,15,16]. In equine dentistry, the risks of spreading blood-borne diseases, such as those that are important in human dentistry [8,24], are not seen as a major risk. In equine dentistry there are, however, risks of spreading various pathogenic and resistant microorganisms [1]. According to the Swedish guidelines on IPC in equine healthcare, dental equipment that comes in contact with mucus membranes, but not penetrating sterile tissue, shall meet criteria for high-level disinfected equipment and surgical dental equipment shall be sterile [10]. According to the American Animal Hospital Associations’ dental care guidelines for dogs and cats, all dental instruments shall be cleaned and sterilized after each use [25] whilst World Small Animal Veterinary Association guidelines states all dental equipment shall be cleaned, disinfected, and/or sterilized based on the equipment’s intended use [26] Also small animal dentistry literature recommends semi-critical and critical instruments to be sterilized after each use [11]. All the recommendations are brief and reflect hygiene recommendations in human dentistry and the differences in guidelines on how to clean, disinfect, and sterilize dental equipment reflects the differences found between countries in human dentistry. All these recommendations are very brief and no clear criteria is given for handpieces. It is important to take into account both the differences and the similarities in risks between human and equine dentistry and to have knowledge of, e.g., the microbial contamination in equine dentistry when developing guidelines on IPC measures in equine dentistry. In this study, equipment and the close patient environment were highly contaminated after dental procedures. The bacterial load in the immediate environment was lower after cleaning and/or disinfecting, but handpieces did not meet the criteria for high-level disinfected equipment after cleaning or disinfecting. 

It can be assumed that a high bacterial load is an indicator of the presence of potentially pathogenic microorganisms. In a study by Adams et al. [27] investigating the occurrence of *Staphylococcus aureus* in a human intensive care unit, most *Staphylococcus aureus* were detected on heavily contaminated hand-touch sites. This study has shown that dental procedures, using handpieces, contaminate the equipment and the surrounding environment, with a potentially increased risk of transmission of pathogens between patients, staff, and facilities. As HSH is used for endodontic treatment, transmission of microorganisms can have severe consequences since the pulp is exposed, i.e., there is an increased risk for infection. To classify HSH as critical equipment could emphasize its importance to improve IPC measures in equine dentistry. For semi-critical equipment as LSH and SH, when not used for endodontic treatment, it can be discussed whether the strictest definition of high-level disinfected equipment is needed in equine dentistry. An important factor for considering to use the less strict definition of high-level disinfected equipment (i.e., free from pathogens and occurrence of occasional vital microorganisms, see Table 1) for semi-critical LSH and SH is that it is important to identify an achievable, and measurable, threshold value for manual cleaning and disinfection. It is difficult to estimate the risk of infection transmission if the less strict Swedish definition of high-level disinfected equipment would be applied for handpieces. The results of our study indicate that even the less strict criterion may require more meticulous routines for cleaning and disinfection.

In this study, the effect of surface disinfection of handpieces was similar to the results reported by Pinto et al. [12] in a study in human dentistry, where handpieces did not meet even the less strict criteria for high-level disinfected equipment in human dentistry [9]. The couplings in both LSH and HSH can be regarded as a bridge between external and internal surfaces. The couplings can, if the IPC measures are ineffective, serve as a vector spreading infectious agents between patients. Infectious agents may occur in horses without clinical symptoms of infection, for example, Pusterla et al. [1] reported respiratory pathogens in 22% of healthy horses submitted for routine dental care.

If the upper limit for bacterial load on external contact surfaces of 2.5 CFU/cm^2^, as suggested in studies on human hospital cleanliness [28,29,30,31], is used as the limit of acceptance for the head support (a non-critical piece of equipment), most samples in this study meet the criteria after cleaning and/or disinfecting. 

Limits of the study: The bacterial load on the shaft of the SH after dental procedures in this study diverges from results reported by Smith et al. [3], in median 1000 CFU/surgical gear compared to up to 30 CFU/shaft in this study. Smith et al. [3] sampled the surgical gear which can be assumed to be more highly contaminated compared to the shaft sampled in this study. Smith et al. [3] also used a better sampling method and a culturing method enabling identification of a wider range of bacteria. If the surgical gear would have been sampled in this study using the same method as Smith et al. [3], it can be assumed the bacterial load would have been considerably higher. Petrifilms ^TM^ were incubated in 30 ± 1 °C which means environmental flora is probably dominating and potential pathogenic bacteria may have been overgrown. The amount of potential pathogen bacteria would probably have been more accurate if Petrifilms ^TM^ had been incubated in 37 ± °C, which is optimal for most mammalian pathogens [32]. Data on length of time in contact with dental tissue was not collected; this data could possibly have provided important information about the degree of contamination after different lengths of time in contact with dental tissue. 

More research will be needed to evaluate how manual cleaning followed by chemical disinfecting of external and internal surfaces of handpieces can result in the less strict Swedish definition of high-level disinfected handpieces, i.e., free from pathogens and occurrence of occasional vital microorganisms. Other topics for future research should be to develop evidence-based guidelines by (1) design and test IPC routines for equine dental procedures and (2) investigate what level of hygiene is needed to minimize the risk of transmission of infectious agents between patients in equine dentistry. 

High contamination of the equipment and the close patient environment, combined with handpieces not meeting criteria for high-level disinfected equipment after manual cleaning or disinfecting, indicates an urgent need for evidence-based guidelines on hygiene in equine dentistry. Based on the study results, protocols for assessment of contamination level could be developed. Larger series of data from several clinics as well as mobile practice should be collected as a basis for such guidelines. Categorization of dental equipment as critical, semi-critical, and non-critical equipment, and a clear definition of hygiene criteria for such equipment are also needed. In addition, guidelines on how to clean, disinfect, and, in some cases, sterilize dental equipment are necessary. Methods for monitoring each step are also needed.

## 5. Conclusions

The detected bacterial load on the equipment and in the close patient environment was often high after dental procedures. Handpieces did not meet the criteria for high-level disinfected equipment after cleaning or disinfecting. In most cases cleaning and/or disinfecting of the head support resulted in a lowered bacterial load. This implies there is a need for evidence-based guidelines on IPC procedures for equine dentistry. In addition, data to support appropriate threshold levels are needed. 

## Figures and Tables

**Figure 1 animals-11-02320-f001:**
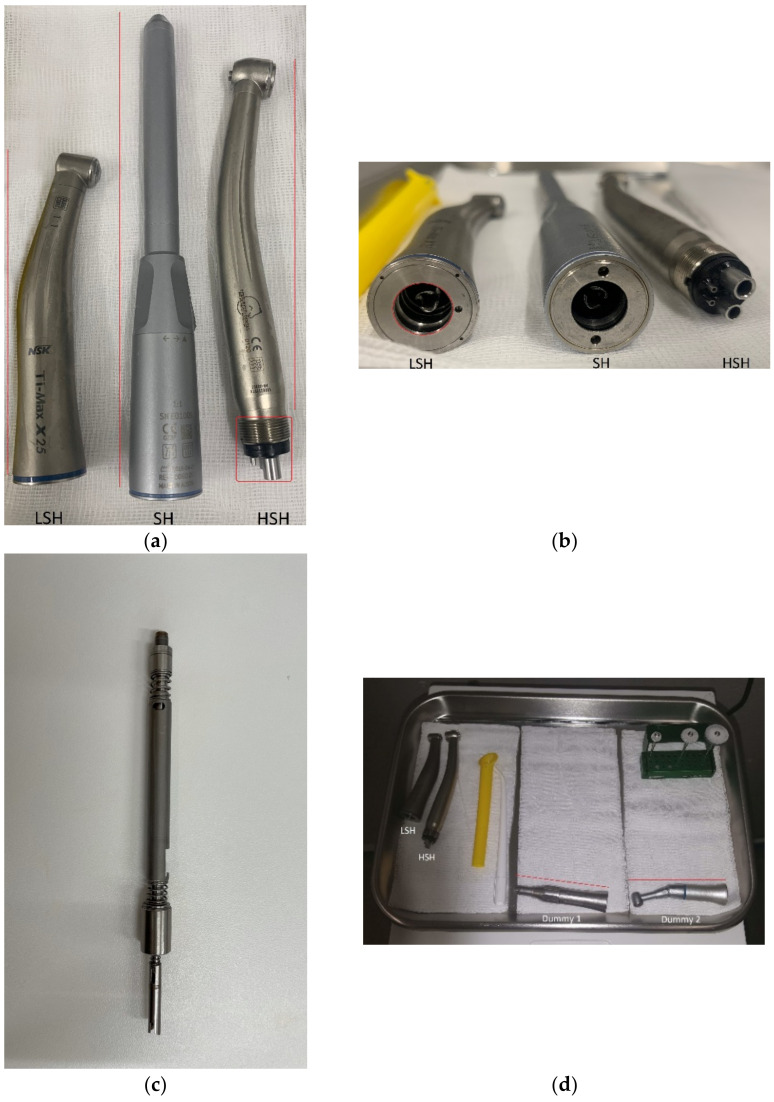
(**a**) Handpieces used for dental procedures in the study. LSH = Low-speed handpiece; SH = Surgical low-speed handpiece; and HSH = High-speed handpiece. Red lines indicate the external surface sampled of LSH, SH, and HSH. The red rectangle indicates the coupling surface sampled on HSH. (**b**) Handpieces used for dental procedures in the study. The red circle indicates the coupling of which the first 0.5 cm was sampled in LSH. (**c**) Shaft of SH; the red line indicates the surface sampled. (**d**) Some of the dental devices used in the study. The red lines indicate the external surface sampled on dummy 1 and 2.

**Figure 2 animals-11-02320-f002:**
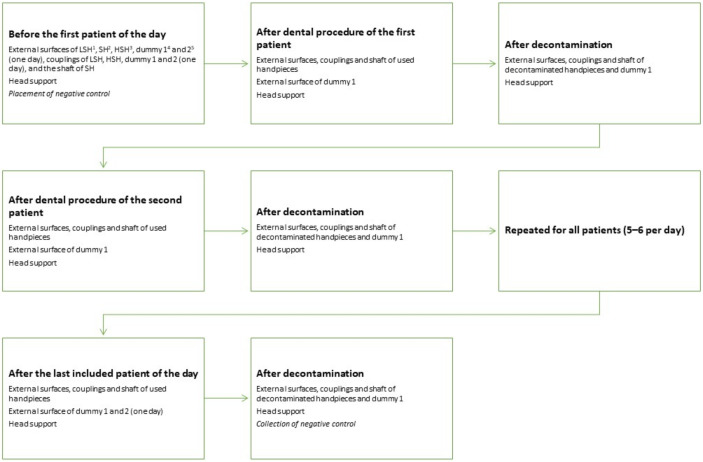
Sampling schedule for the equipment. For assessment of environmental contamination one or two extra LSH (called dummy 1 and 2) and the head support were sampled. The dummies were not used during dental procedure, but consisted of one or two extra handpieces placed close by if needed during the procedure (approximately 70 cm from the patient’s oral cavity). Sampling after dental procedure was carried out within 5 min after the dental procedure was finished and no equipment was rinsed or wiped off before sampling. ^1^ NSK Ti-Max X25, ^2^ W&H S-15, ^3^ Pferdefit-Dental Eco high speed SEA-F4-1-P DEN 1101, ^4^ NSK FX 65, and ^5^ NSK NAC-EC.

**Table 1 animals-11-02320-t001:** Presentation of categorization of dental equipment in human dentistry, area of use, microbial definition according to Swedish guidelines and standards, and the level of cleaning, disinfecting, and sterilizing needed, according to Swedish guidelines.

Category	Use	Example of Equipment Used in Human Dentistry	Example of Equipment Used in Equine Dentistry	Microbial Definition of Category in Swedish Guidelines in Human Dentistry	Level of Cleaning, Disinfecting, and Sterilizing in Swedish Guidelines in Human Dentistry
Non-critical	In contact with intact skin	Spatulas	Mouth specula	Visibly clean [9]	Cleaning if not contaminated; if contaminated, cleaning and disinfecting in a washer-disinfector (WD) or manual cleaning followed by chemical disinfecting [9]
Semi-critical	In contact with mucus membranes, but not penetrating sterile tissue	Handpieces	Drills, burrs and handpieces used when not penetrating pulp, e.g., decreasing enamel ridges	Free from pathogenic micro- organisms and less than one microorganism on 1000 handpieces [9]	Cleaning and disinfecting in a WD [9]
Semi-critical	In contact with mucus membranes, but not penetrating sterile tissue	Compresses	Compresses	Free from pathogenic micro- organisms and occurrence of occasional vital micro- organisms [15]	Cleaning and disinfecting if the equipment is made for reuse
Critical	In contact with sterile tissue	Surgical instruments, like extraction forceps	Drills, burrs, handpieces, and other instruments used when penetrating pulp, e.g., endodontic treatments	Free from living micro- organisms/less than one microorganism on 1,000,000 handpieces [15,16]	Cleaning and disinfecting in a WD followed by sterilizing in a B-autoclave [9]

**Table 2 animals-11-02320-t002:** Description of sampling methods for handpieces and the head support.

Sampling Surface	Sampling Method	Description of Method	Parallel Sampling Method	Description of Parallel Method
External surface LSH ^1^/SH ^2^/HSH ^3^/ dummies	Modified swab sampler method (MSS)	Swabbed once, specification of surfaces (see Figure 1)	NA^4^	NA
Coupling LSH/ dummies	MSS	Swab rotated 360° three times, swabbing possible 0,5 cm into the tunnel (see Figure 1)	NA	NA
Shaft SH	MSS	Shaft swabbed once (see Figure 1)	NA	NA
Coupling HSH	MSS	Swab rotated 360° three times (see Figure 1)	NA	NA
Head support	Swab sampler method [21]	Specified, 10 × 10 cm surface. Swab rubbed three times over area, changing direction, first by 90° then by 45°.	Dip slide	Pressed against surface 15 s, turned and pressed 15 s against an adjacent surface

^1^ Low-speed handpiece. ^2^ Surgical low-speed handpiece. ^3^ High-speed handpiece. ^4^ Not applicable. The swab was not put back in the tube between the three swabbings.

**Table 3 animals-11-02320-t003:** Protocols used for cleaning and disinfection of handpieces and the head support.

Protocol	Equipment	Cleaning and Disinfecting Substance	Description of Cleaning and Disinfecting Methods
1	LSH ^1^, SH ^2^, and HSH ^3^	LD ^4^	Surface disinfection of external surfaces, including external surface of couplings, by rubbing with disinfection wipes (Wet Wipe Triamin Disinfection, Wet Wipes A/S, Vallensbæk, Denmark) until visibly clean for ≥12 s.
2	Head support	LD+EPT ^5^/LD+EIT ^6^	Surface disinfection by rubbing with disinfection wipes until visibly clean for ≥15-s, followed by spraying a surface disinfectant (Dax 75+, KiiltoClean AB, Täby, Sweden or LiV72+, Clemondo, Helsingborg, Sweden) and rubbing the surface for ≥25 s until dry and thereafter spraying surface disinfectant on the surface to air-dry.
3	LSH and HSH	SL-11C+L-10 ^7^	Manual cleaning of external surfaces, including external surface of couplings, by rubbing with cleaning wipes (ICA Städservett, ICA, Solna, Sweden) until visibly clean for ≥12 s.
4	SH	Standard washing liquid+ SL-11C+L-10	Dismantling of SH, manual cleaning with a brush (below the water surface) of SH shell and shaft in warm water until visibly clean for ≥30 s. Rinsing in lukewarm-to-warm water and rubbing with a cleaning wipe until visibly clean for ≥12 s.
5	Head support	SL-11C+L-10 + EPT/EIT	Rubbing with cleaning wipes until visibly clean, for ≥15 s. Followed by spraying a surface disinfectant (Dax 75+, KiiltoClean AB, Täby, Sweden or LiV72+, Clemondo, Helsingborg, Sweden) and rubbing the surface for ≥25 s until dry and thereafter spraying surface disinfectant and left to air-dry.

^1.^ Low-speed handpiece, ^2.^ Surgical low-speed handpiece, ^3.^ High-speed handpiece, ^4.^ Laurylamine Dipropylenediamine, ^5.^ Ethanol, propanol, and tensed, ^6.^ Ethanol, isopropanol, and tensed, and ^7.^ Sodium Laureth-11 Carboxylate and Laureth-10.

**Table 4 animals-11-02320-t004:** Bacterial load of external surfaces, couplings, and the shaft of handpieces after dental procedures and after cleaning or disinfecting. Colony forming units (CFU) is given in total CFU/external surface, CFU/coupling and inner piece, respectively. Pulp exposure is divided into the categories: no exposure, risk for exposure, exposure, and not applicable. The difference between risk for exposure and exposure is that risk for exposure includes dental procedures when the pulp exposure is unintended and exposure includes dental procedures when pulp exposure is intended.

Cleaning or Disinfecting Substance	Type of Handpiece	Pulp Exposure	CFU after Dental Procedure, External Surface	CFU after Cleaning or Disinfecting, External Surface	CFU after Dental Procedure, Coupling or Shaft	CFU after Cleaning or Disinfecting, Coupling or Shaft
LD ^1^	SH ^3^	NA	NA	4 ^7^	NA	0 ^7^
LD	LSH ^4^	NA	NA	8 ^7^	NA	0 ^7^
LD	HSH ^5^	NA	NA	3 ^7^	NA	3 ^7^
LD	SH	No exposure	TNTC ^6^	1	0	1
LD	SH	No exposure	TNTC	28	0	0
LD	SH	Risk for exposure	620	0	2	0
LD	LSH	Risk for exposure	510	1	26	0
LD	HSH	Risk for exposure	820	1420	TNTC	10
LD	SH	Risk for exposure	TNTC	8	1	0
LD	LSH	Risk for exposure	420	1260	20	0
LD	HSH	Risk for exposure	720	70	TNTC	520
LD	SH	No exposure	TNTC	5	1	0
SL-11C+L-10 ^2^	SH	NA	NA	69 ^7^	NA	0 ^7^
SL-11C+L-10	LSH	NA	NA	TNTC ^7^	NA	0 ^7^
SL-11C+L-10	HSH	NA	NA	10 ^7^	NA	8 ^7^
SL-11C+L-10	SH	Risk for exposure	TNTC	TNTC	5	0
SL-11C+L-10	LSH	Risk for exposure	TNTC	231	88	0
SL-11C+L-10	HSH	Risk for exposure	TNTC	TNTC	TNTC	TNTC
SL-11C+L-10	SH	No exposure	TNTC	160	14	30
SL-11C+L-10	SH	No exposure	TNTC	TNTC	4	0
SL-11C+L-10	SH	Risk for exposure	TNTC	25	0	0
SL-11C+L-10	LSH	Risk for exposure	TNTC	900	520	0
SL-11C+L-10	SH	No exposure	TNTC	1350	0	0

^1.^ Laurylamine Dipropylenediamine, ^2.^ Sodium Laureth-11 Carboxylate and Laureth-10, ^3.^ Surgical low-speed handpiece, ^4.^ Low-speed handpiece, ^5.^ High-speed handpiece, ^6^ Too numerous to count, and ^7.^ Samples taken before the first patient of the day.

**Table 5 animals-11-02320-t005:** Bacterial load of dummies (handpieces not used during dental procedure, illustrating extra handpieces placed close by (approximately 70 cm from the patient’s oral cavity) if needed during the procedure. Day 1 dummy 1 was put in place approximately 10 min before the first patient of the day, day 2 dummies 1 and 2 were put in place approximately 10 min before the first patient of the day. Dummy 1 was sampled and cleaned or disinfected after each patient whilst dummy 2 was not cleaned or disinfected, and sampled only after the last patient included for the day. CFU is given in total CFU/external surface and CFU/coupling respectively.

Dummy	Cleaning or Disinfection Substance	CFU after Dental Procedure, External Surface	CFU after Cleaning or Disinfection, External Surface	CFU after Cleaning or Disinfection, Couplings
1	NA	NA	183 ^3^	22 ^3^
1	LD ^1^	TNTC	1	0
1	LD	1340	18	3
1	LD	0	0	0
1	LD	67	0	0
1	LD	82	0	0
1	LD	1100	8	0
1	LD	NA	440 ^3^	1 ^3^
2	NA	NA	267 ^3^	0 ^3^
1	SL-11C+L-10 ^2^	TNTC	0	0
1	SL-11C+L-10	330	2	1
1	SL-11C+L-10	87	1	0
1	SL-11C+L-10	18	0	0
1	SL-11C+L-10	35	0	0
2	NA	TNTC	NA	NA

^1.^ Laurylamine Dipropylenediamine, ^2.^ Sodium Laureth-11 Carboxylate and Laureth-10, and ^3.^ Samples taken before the first patient of the day.

**Table 6 animals-11-02320-t006:** Bacterial load detected on the head support after dental procedures, and after cleaning and/or disinfecting. Colony forming units (CFU) are given in CFU/cm^2^. For bacteriological analyses 3M Petrifilm ^TM^ aerobic count (AC) Plates and dip slides with TSA agar/TSA agar with a neutralizer, neutralizing several disinfectants.

Cleaning and Disinfecting Substances	Bacterial Load after Dental Procedure			Bacterial Load after Cleaning and/or Disinfecting		
	Petrifilm ^TM^	Dip Slide TSA + Neutralizer	Dip Slide TSA	Petrifilm ^TM^	Dip Slide TSA + Neutralizer	Dip Slide TSA
LD^1^+EPT^2^/LD+EIT ^3^	NA	NA	NA	0.03 ^5^	0.11 ^5^	0 ^5^
LD+EPT/LD+EIT	TNTC	7.23	TNTC	3	0.11	0.96
LD+EPT/LD+EIT	TNTC	TNTC	TNTC	0.04	0.11	0
LD+EPT/LD+EIT	TNTC	TNTC	TNTC	0.07	0	0
LD+EPT/LD+EIT	TNTC	5.32	9.89	0.06	0	0.11
LD+EPT/LD+EIT	TNTC	1.49	6.38	0	0	0
LD+EPT/LD+EIT	TNTC	TNTC	TNTC	0.01	0	0.85
SL-11C+L-10 ^4^ + EPT/EIT	NA	NA	NA	TNTC ^5^	NA	NA
SL-11C+L-10 + EPT/EIT	TNTC	NA	NA	1.45	NA	NA
SL-11C+L-10 + EPT/EIT	TNTC	NA	NA	0.10	NA	NA
SL-11C+L-10 + EPT/EIT	TNTC	NA	NA	TNTC	NA	NA
SL-11C+L-10 + EPT/EIT	TNTC	NA	NA	0	NA	NA
SL-11C+L-10 + EPT/EIT	TNTC	NA	NA	0.73	NA	NA

^1.^ Laurylamine Dipropylenediamine, ^2.^ Ethanol, propanol and tensid, ^3.^ Ethanol, isopropanol and tensid, ^4.^ Sodium Laureth-11 Carboxylate and Laureth-10, and ^5.^ Samples taken before the first patient of the day.

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
