# Peer review of "Bacterial Contamination of Equine Dentistry Equipment—Effect of Cleaning and Disinfection"

_animals, 2021, doi:10.3390/ani11082320_

Round 1
Reviewer 1 Report
- Key words. Suggest to make 5 key words only and delete cleanliness.
- To consider use in introduction small animal dentistry literature available:
Terpak, C. H. & Verstraete, F. J. M. (2012) Instrumentation, Patient positioning, and aseptic technique. In: Oral and maxillofacial surgery in dogs and cats. Eds F. J. M. Verstraete and M. J. Lomer. Saunders Elsvier, St. Louis. pp 55-68
-
Page 4 line 123 it should be (c) instead of (b) I suppose,
Can you comment please what is foamy material present on the surface?
- Page 5 line 142
were the gloves sterile?
- Page 6 line 160. Collecting samples before the first patient was performed after the most recent use which could be various depending on when the most recent use happened. Can you add information if it was after 1-2 days (weekend) or just the day after.
- Page 6 line 170.
There could be information about storage of handpieces after cleaning: were they isolated, packed into sterile bags.
- Page 10 line 259-260. That does not refer to the situation when pulp is already necrotic and infected.
Author Response
Thanks for your helpful comments. Please see the attachment.

Reviewer 2 Report
Line 48 - "in human dentistry human dental equipment" would read more correctly with deletion of "human dental equipment"
Line 88 - "equine dental equipment are rarely" should read "equine dental equipment is rarely"
Author Response

(The authors gave the same response as above.)

Reviewer 3 Report
A fundamental issue for this paper is to define exactly what type of dental procedures are being performed here. It may be that practice is a bit different in Sweden but generally dental procedures can be dived up into “routine” (generally not exposing pulp chambers and not using the types of handpieces mentioned) and more advanced procedures (including endodontic procedures). The advanced procedures are far less commonly done. Perhaps the “routine” procedures are not done in this clinic? Perhaps they are done by not veterinarians?
65: These references are relevant to aerosol spread of normal flora bacterial in the mouth rather than contamination from the devices.
88: Although it is not defined by Pusterla, this is confusing two very different procedures. “Routine” equine dental procedures is using predominately hand rasps (or sometimes motorized rasps) and involves mainly decreasing enamel ridges on the edges of the dental arcades - not exposing pulp chambers. It doesn’t entail the use of the type of handpieces you have mentioned. Therefore the expected level of instrument preparation is considered to not require sterility, but good hygiene and disinfection (semi-critical).
90: Post-graduate specialist equine dental courses for veterinarians would not be expected to have much detail on disinfection and sterilization – it would be expected that it is covered in the veterinary qualification.
94: I can’t read Swedish but my intuition from scanning https://svf.se/media/10mbpucp/infektionskontroll_inom_hastsjukvard.pdf is that sterilisation of instruments that equipment in normal use penetrates skin or mucous membrane (which I would assume includes exposure of the pulp chamber) requires sterilization (p45 Tabell2).
99: Where is the reason to conclude the use of hand pieces that expose pulp cavities are “semi-critical”?
118: As the contamination potential of different handpieces may vary, more specific detail of the make and model of them should be given. Water-cooled – what flow rate? More specific information should be given on typical use e.g length of time in contact with dental tissue.
123: Figure 1 c is mislabeled
138: More detail on the swabbing of the handpieces need to be given. How long after coming out of the mouth? Was there any rinsing or wiping of the handpieces done before sampling? The effect of residual organic matter moisture or saliva is likely to effect the bacterial counts.
148: An abbreviated version of the appendix Table 2 should be included in the actual paper. The abbreviations in Table 2 should be specifically given rather than alluded to via superscripts.
149: specify what is meant as the “head-support”. Is this a gag? What type. Give specific detail on exactly where the swabbing took place. If it is an area likely to be contaminated by fluid coming from the mouth then that would change the expected contamination.
172: Detail the surface disinfection process – emersion / spraying drying afterwards?
204: Detail the type and degree of pulp exposure made.
206 / 226: Define Too Numerous To Count - was there an attempted count? In Discussion you might want to suggest how this could be better handled.
243: Blood borne human diseases are not relevant to this situation, but there are blood-born diseases that can spreads from horse to horses (and even horse to human) that are. Even more concerning is the greatest percentage of cases involved pupil exposure and non-sterile handsets are being used. The list of organisms is far greater than the respiratory organisms mentioned by Pusterla.
244: If you think this is a justified conclusion reference (and quote) the “Swedish Guidelines” specific to this. I do not believe Swedish human dentists would have guidelines stating that “hygiene measures” are suitable for hand pieces. According to Table 1: “semi-critical one microorganism per 1,000 handpieces / critical one in 1,000,000 . Also reference standard requirements in veterinary dentistry from other parts of the world if you are wanting to make a point about this.
251: it is unclear what you are implying by this. Is it really relevant?
261: but your data indicates that LSH and SH handpieces are also used in endodontic work.
265: the precise Swedish definition needs to be given to evaluate this. Some of your results appear to more than occasional microorganisms – this statement is very loose.
- The “less strict definition” appears very achievable. This statement doesn’t make any clear conclusion.
271: Completely disagree – Pinto refers to the pre-HIV practice of soaking ion alcohol for some time – considerably better than using disinfectant wipes. I don’t believe it is comparable.
276: you haven’t shown a link between respiratory pathogens and contamination of couplings – what are you trying to say?
279: These references refer to general surfaces – be clear you are not implying this is suitable for instruments.
295: Wiping with a disinfectant-soaked wipe has a different outcome to soaking in disinfectant. If the “disinfecting” process you have described leads to longer than fleeting exposure to the disinfectants you would need to explain how.
Author Response

(The authors gave the same response as above.)
